# Laws and Trends of the Evolution of Traditional Villages in Plane Pattern

**Xi Yang [1],*** , **Ke Song [1] and Fuan Pu [2]**

[1]    School of Architecture, Harbin Institute of Technology (Shenzhen), Shenzhen 518055, China; songke@hit.edu.cn

[2]    School of Software, Tsinghua University, Beijing 100084, China; pfa12@tsinghua.org.cn

*    Correspondence: xi-yang12@tsinghua.org.cn; Tel.: +86-134-6659-9521

**Abstract:** This study collected and analyzed dynamic spatial data of eight traditional villages scattered in different regions of China. A multi-temporal analysis of morphological metrics of spatial patterns and a regression analysis of the morphological evolution were used to analyze and contrast the historical spatial processes of different villages. These were then compared using patch texture and rural macro-morphology perspectives. This led to an assessment of the general laws and trends associated with rural spatial processes. (1) There has been a significant shift in the stability of rural spatial development since the founding of the People's Republic of China (PRC). (2) Most small and medium-sized villages have maintained a relatively stable spatial texture, while large villages have changed significantly. (3) The mean and variance of the patch area, and the Euclidean nearest-neighbor distance, are correlated in some cases. (4) The mode of rural expansion may be relevant to limitations in the total area of growth. (5) The fractal dimension of the rural macro-morphology may follow a morphological order of oscillation around the equilibrium level. (6) The common mean value of the projected area of rural building patches is expected to be 100 m$^2$.

**Keywords:** traditional rural settlements; dynamic evolution; patch texture; macro-morphology; multi-temporal analysis; morphological metrics

---

## 1. Introduction

The layout of traditional rural residential land constitutes the core framework of the rural cultural landscape. The evolution of that land is the direct manifestation of the interaction between humans and their environment [1]. Theoretically, this process is an iterative cycle of "spatial form shaping" and "environmental feedback". During the rapid urbanization of modern China, many rounds of spatial shaping may have occurred before the environmental feedback. This has resulted in a nonlinear relationship between human activities and environmental constraints [1]. Human activities sometimes advance more rapidly than environmental feedback. Therefore, planners need to acquire more knowledge about the spatial laws and reasonable expectations with respect to dynamic trends. This is needed to integrate natural and cultural rationality into spatial development, in the absence of immediate environmental feedback (Figure 1).

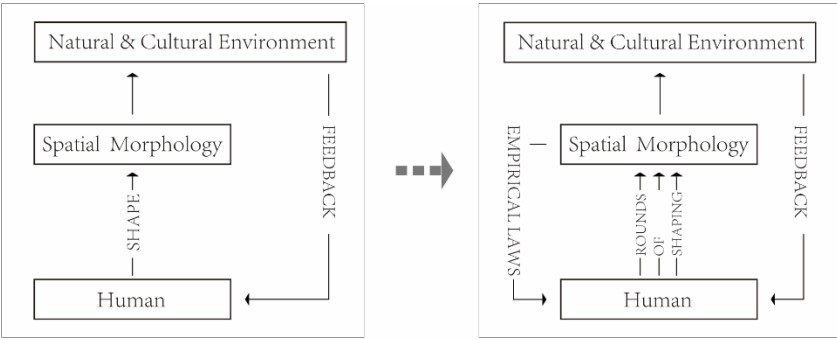

**Figure 1.** The changes in human–environment relationship.

The spatial morphology of villages is a geographical phenomenon. It is important to grasp the spatiotemporal dynamic process of the geographical phenomenon, rather than simply obtaining the spatial pattern once formed. Combining the two categories of time and space into a unified dynamic system is needed to truly understand the essential laws of the spatial morphological development [2].

As such, it is important to evolve from considering rural space as a static form to considering it as a dynamic system. This would more effectively consider the dynamic non-equilibrium process through which the spatial system reaches the equilibrium state. For a specific village, to guide an appropriate spatial decision-making before the intervention of spatial planning, analysis, and judgment are needed to consider the following questions: (1) Did specific laws shape the historical process of spatial evolution? Did these rules change during the process? (2) Has the current rural spatial form reached a relatively stable state? If not, how does the existing logic inform development trends? (3) Has the current trend continued the macro trend seen with historical processes?

There are two methods for analyzing spatiotemporal dynamics:

The first method is spatiotemporal dynamic modeling. This method usually creates a model rooted in an intelligent algorithm, building a mathematical relationship between the constraint conditions and the morphological evolution [3,4]. Examples include the cellular automata model and the system dynamic model. These deduce spatial trends from previous spatial laws.

In general, spatiotemporal dynamic models can be divided into two types, based on the embedded algorithms. One type is the black box algorithm (e.g., artificial neural network algorithm), which maps input features to an output target simply based on the data. The second type is a transparent algorithm (e.g., decision tree algorithm and Bayesian algorithm), which learns the mapping based on assumptions and data [5]. When applying models for spatial analysis and prediction, it is not enough to determine the final answer (the "what"). It is also important to generate a reasonable explanation about how the answer was reached (the "why") [6]. That means that interpretable transparent models generally match spatial research needs and perform better than black box models, if there are good assumptions based on credible experiential knowledge and rules [7,8]. In this case, it is necessary to analyze and distill experiential knowledge from historical spatial processes.

The second method involves morphological metrics, which quantify, interpret and assess complex spatial structures. These metrics reveal spatial system properties that are not easy to directly observe, using a working platform called Fragstats [9,10]. Static morphological metrics are generally referred to as landscape metrics [11]. These metrics are used in landscape ecology, and are also called spatial metrics [12]. The group of metrics is an important tool for analyzing mapping products with a plane configuration. This provides an additional level of quantitative information about the structure and pattern of the space, and these metrics have dominated studies that apply quantitative analyses to urban forms for some time [13]. In fact, the application of morphological metrics in spatial analysis occurred earlier than the application of the spatiotemporal dynamic model. Morphological metrics can provide an empirical basis of knowledge [14,15] and rules [16] for the creation and evaluation of spatiotemporal dynamic models.

However, static morphological metrics merely describe the spatial patches in a single time point. They do not reflect dynamic information about the spatial evolution. Because of this, previous research has tried to empower the metrics, with the ability to quantitatively characterize the spatial change process [17]:

(1) The changes for a specific place were shown using the time-series data of static morphological metrics [17–20]. (2) The Landscape Expansion Index (LEI) was used to analyze and define the types of spatial expansion between different phases [21,22], including infilling, edge-expansion, and outlying modes. However, the LEI index has some problems recognizing expansion patterns when examining patches with special shapes. Scholars have improved this index, or have proposed alternative indexes, to strengthen the accuracy of spatial recognition [23–26]. Auxiliary indices have also elaborated descriptions of the spatial process [27].

For morphological metrics, when studying the spatial evolution of villages, the LEI may be no better than the time series analysis with multi-temporal metrics. First, the continuous spatial evolution should be described using the rules of spatial transition between different steps or throughout the full spatial process. Based on the essential properties of LEI, the metrics describe the "differences" between different spatial patterns of construction land across different developing phases, not the rules or trends of those "differences". This suggests there is no essential difference between the LEI and static morphological metrics. Second, when the spatial scale for research is narrowed to a single village, building patches are often in a scattered layout, expanding in multiple directions. This makes it easy to misjudge the types of spatial expansion when measured by LEI. Third, from a research method perspective, the spatial scale for a particular village is smaller than for a city. Therefore, its evolution mode is relatively stable, making the spatial expansion discernable by the naked eye.

In addition, due to the difficulty of data acquisition, many previous studies on dynamic spatial evolutionary morphological metrics have focused on comparing and analyzing data for two time-points. Many possible results could be extrapolated from such a one-step difference. As such, it is difficult to infer a reliable trend of changes in the data. It may be possible to hypothesize whether the value will go up or down, but impossible to deduce the definite value (if there is one) towards which the index value will trend. Therefore, more spatiotemporal data are needed. Additionally, absolute values of morphological metrics depend on the spatial resolution and the extent of the study area. In other words, a change in the extent of the study area or in the spatial resolution will significantly change the metrics. This limits the ability to relate the results of one region to another [17]. Hence, because it extends beyond a single datum, trend analyses are less impacted by the problems above, and are a better fit for this study.

Using eight villages in China as research samples, this study collected and analyzed spatial evolution data. The study then applied a set of morphological metrics to conduct a multi-temporal analysis of the spatial evolution characteristics, from the perspective of patch texture and rural macro-morphology. Based on that, the general characteristics and development trends of traditional rural spatial dynamics were compared and summarized, providing essential empirical knowledge to support spatiotemporal dynamic modeling and predictions related to rural evolution.

## 2. Data and Methods

### 2.1. Data of Rural Evolution

Few historical spatial data related to traditional villages in China are available, and the corresponding reconstruction of the rural spatial process remains in the initial stage. Therefore, it is difficult to capture direct data related to rural evolution. Based on our former work of rural data reconstruction as well as the work of other rural researchers, we selected eight traditional villages distributed across China as the study sample: ShangZhuang Village (in Shanxi Province) [28], Doushan Village (in Jiangsu Province) [29], Xidi Village (in Anhui Province) [30], Bailu Village (in Jiangxi Province) [31], Tanshui Village (Fujian Province), QuanFu Village (in Yunnan Province) [32],

Qiaoxiang Village (in Guangdong Province) [33], and Kengzi Village (in Guangdong Province) [34] (Figure 2). The spatial data were then processed into multitemporal maps. There were two main conditions driving the reconstruction of the spatial evolution. First, for some sample villages, the related references provided accurate spatiotemporal data for direct use. Second, for others, the database provided the contemporary spatial patterns of the settlements, supplemented by the historical records of the buildings. This allowed for the reconstruction of the spatial process. Those sample villages had clear differences with respect to spatial scale, the temporal scale of historical development, and the mode of evolution (Table 1). All of the sample villages are natural villages in the same administrative level, and are listed in the order of their geographical location from north to south. The scale differences of those villages indicate the increasing degree of clan organization from north to south, which is consistent with the social reality of China [35].

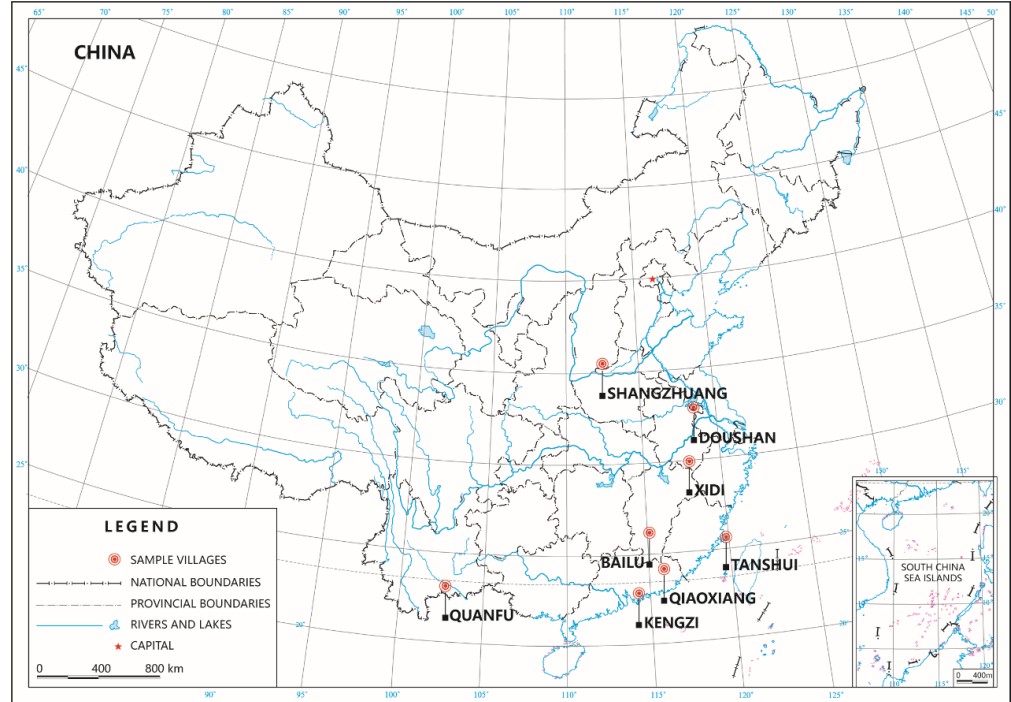

**Figure 2.** Locations of sample villages.

**Table 1.** Characteristics of the sample villages.

| Village | Scale of Its Smallest Bounding Rectangle | Year of Establishment | Spatial Expanding Mode |
|---|---|---|---|
| Shangzhuang | 516 m × 451 m | About 1470s | E+I |
| Doushan | 445 m × 300 m | About 1890s | E |
| Xidi | 681 m × 264 m | About 1050s | E+I |
| Bailu | 1315 m × 882 m | About 1150s | E |
| Tanshui | 1322 m × 676 m | About 1500s | O+I |
| Quanfu | 1565 m × 567 m | About 1010s | O+E |
| Qiaoxiang | 2527 m × 804 m | About 1500s | O+I |
| Kengzi | 5949 m × 4311 m | About 1690s | O |

Note: I: infilling; E: edge-expansion; O: outlying mode of expansion.

The selection of sample villages was random with respect to location and history, but strict with respect to village data reliability. The preliminary analysis did not reveal clear rules in the difference of the founding time of those sample villages from northern China to southern China. However, in the gradually growing rural area, there was a clear change in the spatial expansion mode from edge-expansion to outlying expansion.

*2.2. Data Processing*

Data processing was completed in four steps (Figure 3):

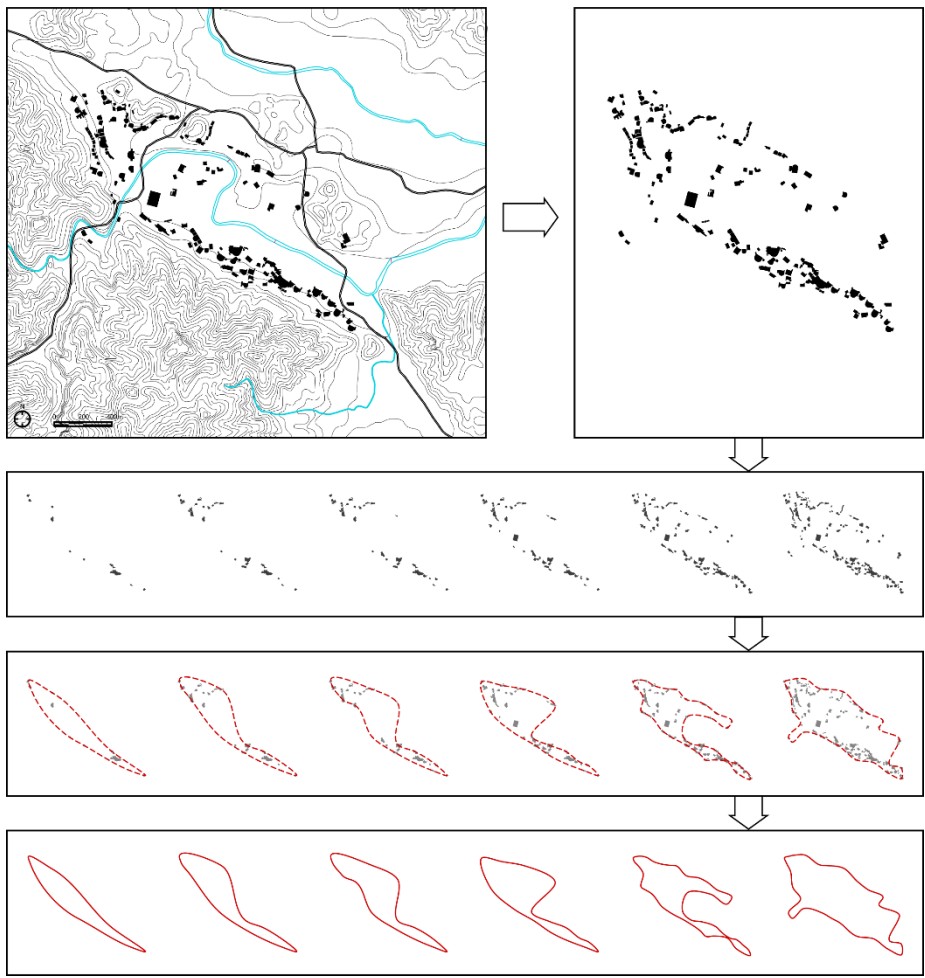

**Figure 3.** Data processing.

(1)  Village map registration was completed by referring to satellite remote sensing images. Most of
     the sample village maps were highly accurate with small deviations, because of achievements in
     field surveying and mapping.
(2)  The data for residential land patches in each map were extracted.
(3)  The multi-temporal layouts of the residential land patches in each village were reconstructed,
     using records in the corresponding village studies.
(4)  The boundaries for the village layout in each development phase were generated using data
     management tools and cartography tools in ArcMap. To avoid the influence of the specific form
     of those buildings on the boundary generation, for a certain village, the gravity center of each
     building plaque was extracted. This constituted the point cluster of the village. Then, the outer
     boundary of the point cluster was generated using ArcMap; this was guided by the principle that
     any line segment connecting two points on the boundary should be shorter than the threshold
     value (which is accurate to the meter). The threshold value was the minimum value needed to have
     the boundary enclose all points. Finally, the boundary, expressed by a polyline, was softened using
     the smoothing algorithm of Bezier Interpolation, embedded in the Smooth Line Tool of ArcMap.
     When the fitting curve intersected, the threshold value was reduced, and multi-boundaries were
     regenerated to enclose a multi-cluster of points (Figure 4).

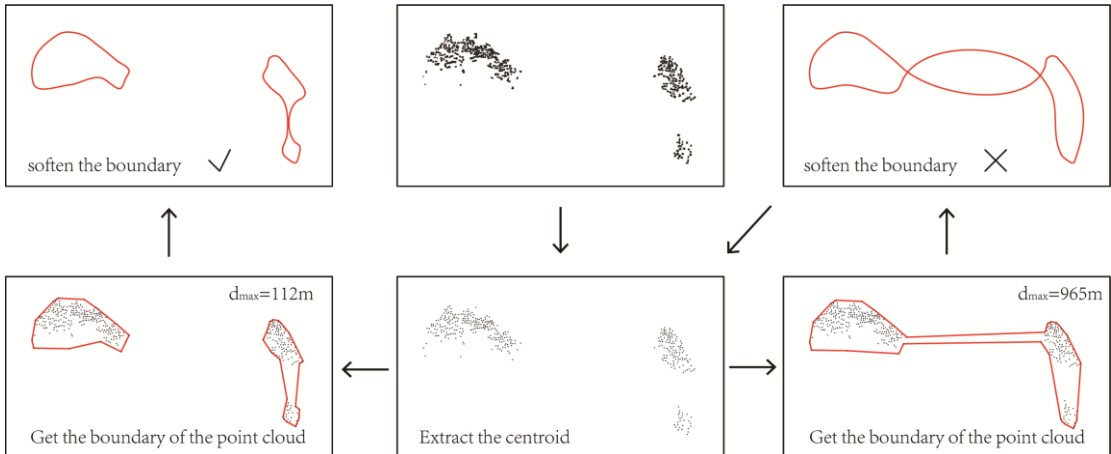

**Figure 4.** Treatment for special boundaries.

## 2.3. Methods for Spatial Analysis

This study analyzed the rural spatial evolution using the patch texture and rural macro-morphology. We introduced mean and standard deviation of patch area and Euclidean nearest-neighbor, as well as distance and patch density at the patch texture level, and area and fractal dimension index at the macro-morphology level. These served as qualitative metrics to compare the morphological evolutional difference and explore the common rules of different villages.

Based on information redundancy in existing morphological metrics [36,37], the options of metrics for each study level, covering the size, shape, and arrangement of patches, were narrowed to be as concise as possible. Prerequisites for use included a relatively direct guidance for spatial practice, and qualitative indicators that are not easy to obtain through observation. The presence of a weak correlation among different groups was considered while screening out the metrics (Table 2).

**Table 2.** Spatial metrics selected in this study.

| Metrics | | | Abbreviation | Description | Unit |
|---|---|---|---|---|---|
| Spatial texture | Patch Area | Mean | AREA_MN | The average area of all patches in the same category | Hectares |
| | | Standard Deviation | AREA_SD | Standard deviation of the area of all patches of the same class | Hectares |
| | Euclidean Nearest-Neighbor Distance | Mean | ENN_MN | The average minimum distance between the individual patches. | Meters |
| | | Standard Deviation | ENN_SD | Standard deviation of the minimum distance between each individual patches of the same class | Meters |
| | Patch density | | PD | The number of patches of the corresponding patch type divided by total landscape area, which facilitates comparisons among landscapes of varying size | Number per 100 hectares |
| Integral shape | Area | | AREA | The patch area, which is the basis for many other indices | Hectares |
| | Fractal Dimension Index | | FRAC | The fractal dimension describes the complexity and the fragmentation of a patch using a perimeter–area proportion. Fractal dimension values range between 1 and 2. Low values were derived when a patch had a compact rectangular form with a relatively small perimeter relative to the area. More complex and fragmented patches were associated with an increased perimeter and yielded a higher fractal dimension. | —— |

Finally, we isolated any abnormal values in the multi-temporal morphological metrics of each sample village, and analyzed the causes for appropriate data filtering. Then, we completed a regression analysis on the changes in the data related to morphological metrics, to delineate similarities and differences among different types of villages.

## 3. Results and Analysis

### 3.1. The Morphological Evolution of the Sample Villages

Figure 5 shows the morphological evolution of the sample villages, with different scales for different villages. Rural settlements are expressed as black patches enclosed within a red-colored boundary marking the village. This allowed a comparison of the sample villages with similar identifiability within the same timeline.

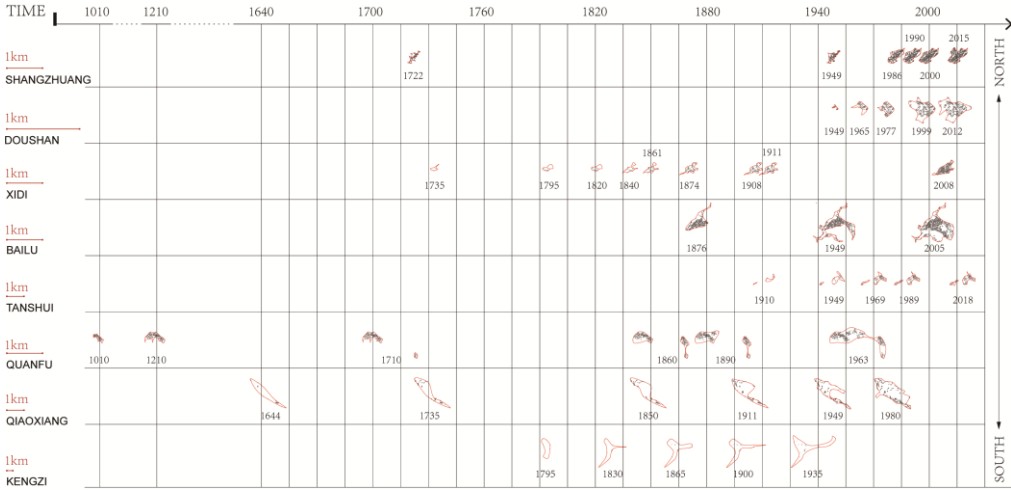

**Figure 5.** Spatial evolution of sample villages.

The spatial process reconstruction of the sample villages illustrates some differences between Northern and Southern China. First, the earliest traceable spatial records for Southern China are from an earlier time period than for Northern China. By contrast, the Chinese population historically migrated from north to south. The discrepancy may relate to the differences between the two macro zones with respect to the size and power of clan groups. The clans in southern China had a very large population and economy [38]. Therefore, their village chronicles and clan history were recorded and preserved in good condition in oral or written form. Second, compared with the northern villages, the southern villages experienced a lower degree of spatial compactness and concentricity. This may be related to the topography and natural resource distribution [38]. Third, the speed of rural spatial aggregation tended to gradually decrease from north to south. Specifics were then further analyzed using morphological metrics.

### 3.2. The Metric Data Changes in the Spatial Sprawl

The morphological metrics were individually computed for each of the eight sample villages and for the year, using corresponding spatial records.

3.2.1. The Data Changes of the Metrics about Patch Texture

(1) Patch Area Mean (AREA_MN).
This metric reflects the size of the rural family unit. Figure 6a shows the quantitative historical change in the metrics for the eight villages. Qiaoxiang, Shangzhuang, and Kengzi were associated with a large range of data; as such, the data for the other five villages are shown more clearly in Figure 6b.

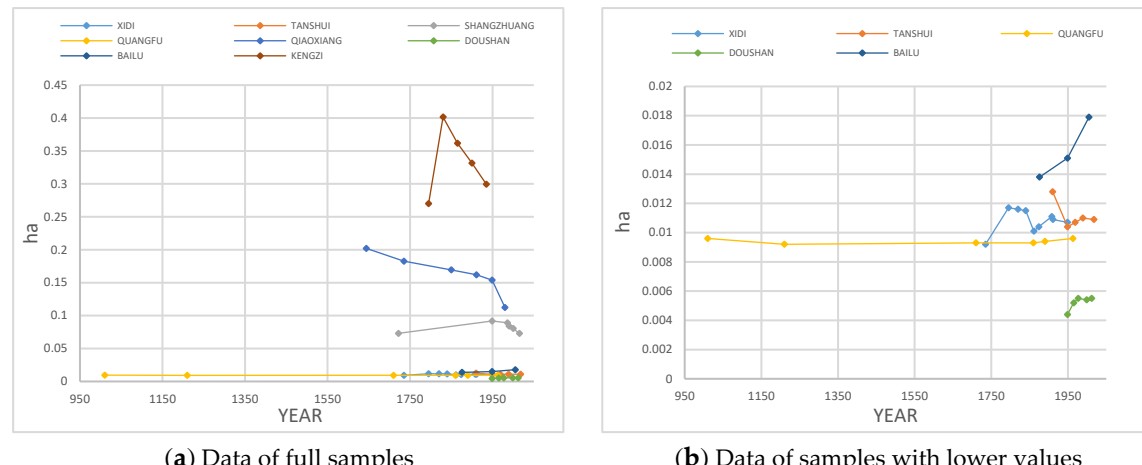

(**a**) Data of full samples         (**b**) Data of samples with lower values

**Figure 6.** Variation diagram of Patch Area Mean.

The data for the average patch area of most villages, with the exception of Kengzi, Qiaoxiang, and Shangzhuang, fluctuated within a 50–150 m$^2$ range, with a concentration around 100 m$^2$. The data changes in the spatial process of the five villages were relatively subtle and essentially remained stable, with the exception of Bailu village, which experienced an upward trend.

For Kengzi, Qiaoxiang, and Shangzhuang, the average patch area occurred at a higher order of magnitude. The data for Qiaoxiang show a downward trend. By contrast, the data of Kengzi and Shangzhuang showed an initial increase, with a subsequent decrease. There was no significant difference between the values at the rising start and the falling end.

The data for Qiaoxiang and Shangzhuang showed a rapid excessive downward trend after 1949. Using the Qiaoxiang data as an example, a logistic regression model was used to fit the change in the data, without including the abnormal sampling point of 1980 (Figure 7). The projected trend indicated it would take approximately 380 years from the fifth time point if the average patch area decreases to the value equal to that of the sixth time point in reality. However, this process required just 30 years, demonstrating that the size of the family unit in some villages inhabited by large clans shrank abnormally since the founding of the People's Republic of China (PRC).

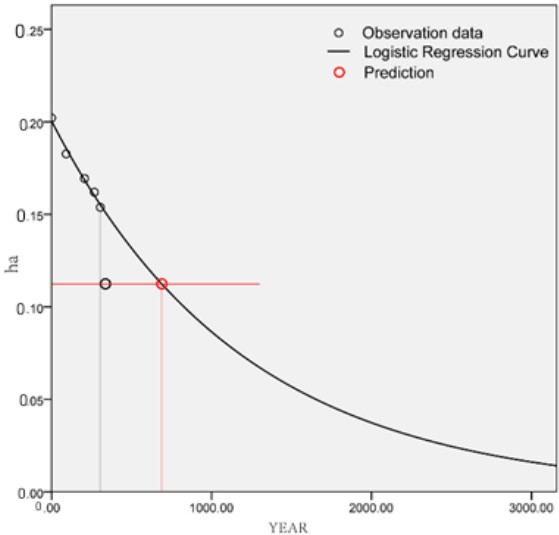

**Figure 7.** Trend analysis of AREA_MN for Qiaoxiang Village.

(2) Patch Area Standard Deviation (AREA_SD).
This metric reflects the degree of uniformity in patch size.

The data changes shown in Figure 8 indicate that AREA_SD experienced a slight upward trend when the corresponding AREA_MN was small (less than 500 m$^2$). The AREA_SD experienced a significant downward trend when the corresponding AREA_MN was large (greater than 500 m$^2$). In other words, villages with small dwelling units had a more stable uniform patch size. The patch-size difference in villages with large dwelling units significantly weakened as time passed.

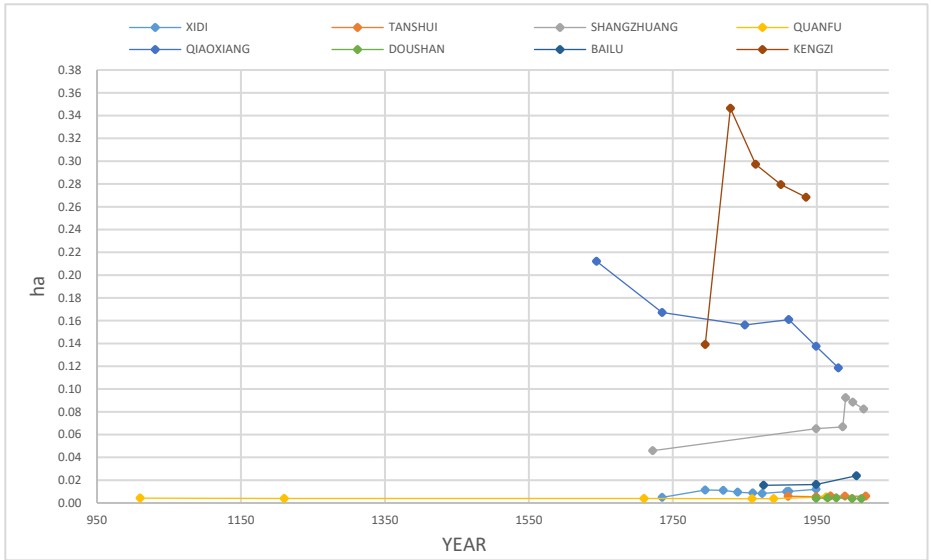

**Figure 8.** Variation diagram of Patch Area Standard Deviation.

Consistent with the changes in the AREA_MN values, the values associated with the last few points for Shangzhuang and Qiaoxiang with respect to these metrics experienced either an abrupt rise or a decline.

For AREA_MN and AREA_SD, Table 3 provides the correlation evaluation of the two metrics in each village. These two metrics show a highly significant correlation in Xidi, Qiaoxiang, and Kengzi, where the value of AREA_SD was proportionately related with AREA_MN (Figure 9). In other words, the coefficient of the standard deviation of AREA (patch) was stable. Both Qiaoxiang and Kengzi are Hakka villages in Guangdong Province. "Hakka" means "guest families" in Chinese, and it represents a subclass of the Han majority in China. The Hakka's ancestors may have arrived from central China centuries ago. In a series of migrations, they settled in their current locations in south China, and have since lived with four generations or more in a large enclosed house. The stability of the coefficient of the standard deviation of AREA (patch) may be related to rural family size.

**Table 3.** Test of the significance of correlation between AREA_MN and AREA_SD.

|  | Shangzhuang | Doushan | Xidi | Bailu | Tanshui | Quanfu | Qiaoxiang | Kengzi |
|---|---|---|---|---|---|---|---|---|
| **Significance level-P (Valid: P < 0.05)** | 0.881 | 0.912 | 0.007 | 0.153 | 0.757 | 0.081 | 0.007 | 0.037 |
| **AREA_SD/AREA_MN** | – | – | 0.93 | – | – | – | 0.97 | 0.86 |

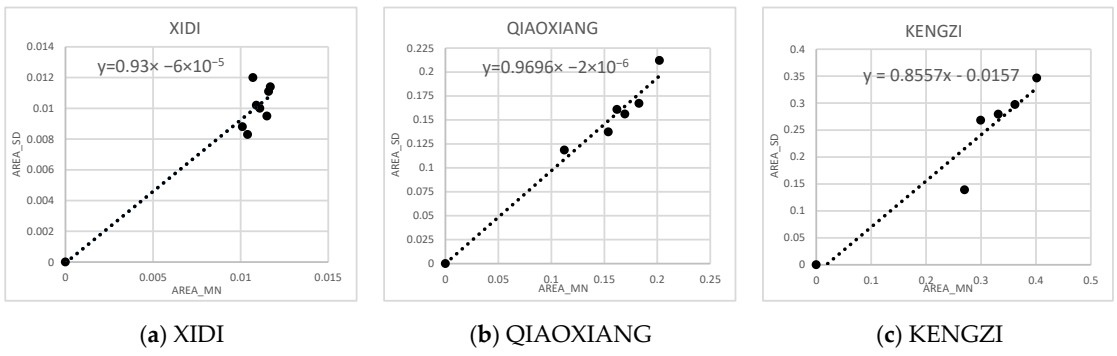

**Figure 9.** Linear regression for AREA_MN and AREA_SD.

(3) Euclidean Nearest-Neighbor Distance Mean (ENN_MN).

These metrics can effectively reflect the degree of patch aggregation [39]. The scatter diagram (Figure 10) shows that the changes in the distances between buildings were relatively slight in most villages, remaining between 2.5 m and 5.5 m. However, the changes in the values were evident in Xidi, Qiaoxiang, and Kengzi, which all showed a consistently decreasing trend. The starting ENN_MN values for Xidi and Qiaoxiang were very high and their steady values were not significantly different from the value in other agglomerating villages. From this perspective, the most distinctive village was Kengzi, because there was an apparently high threshold value in the reduction of ENN_MN.

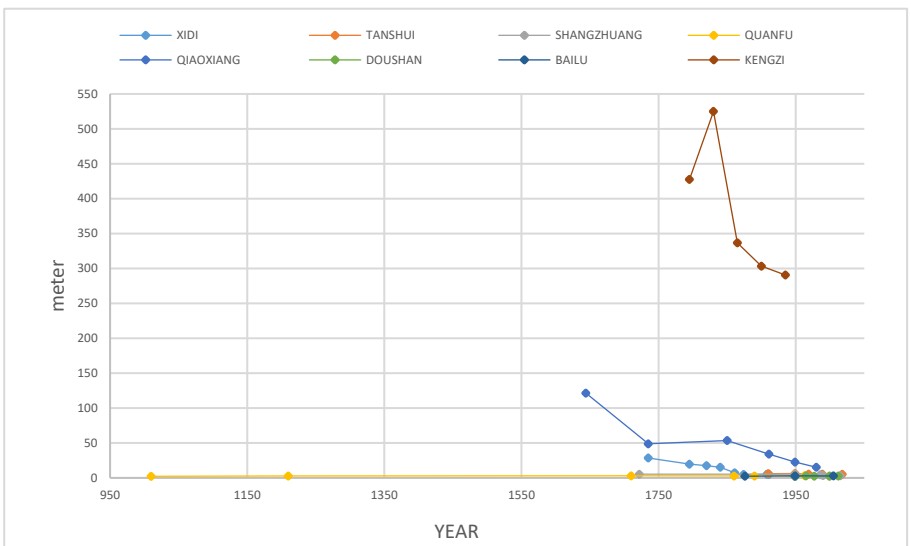

**Figure 10.** Variation diagram of ENN_MN.

The developing trend of Kengzi-ENN_MN was best represented as an inverse S-Curve, omitting one abnormal sampling point (1830) (Figure 11). The value stabilized at 230 m.

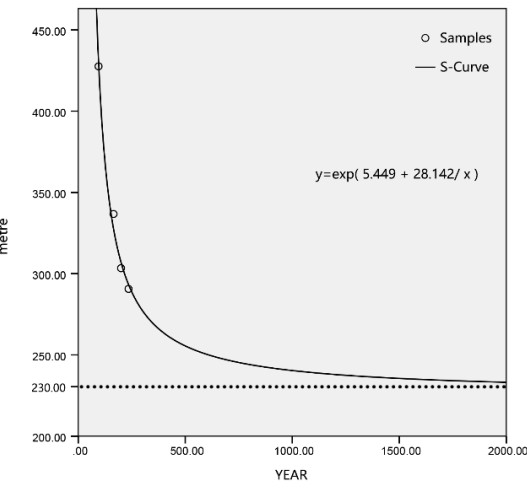

**Figure 11.** Trend analysis of ENN_MN for Kengzi Village.

In addition, villages with a large AREA_MN value also had a large ENN_MN VALUE in the early development stage. This feature became less pronounced as time passed.

(4) Euclidean Nearest-Neighbor Distance Standard Deviation (ENN_SD).

This index reflects the degree of uniformity in the distance between patches (Figure 12). The change in the ENN_SD data indicated a similar trend as with the ENN_MN data. There was a particularly significant correlation between the two metrics in Shangzhuang, Xidi, and Quanfu. For these three villages, the value of ENN_SD was proportionally related to the ENN_MN value. In other words, the coefficient of the standard deviation of ENN was stable (Table 4) (Figure 13).

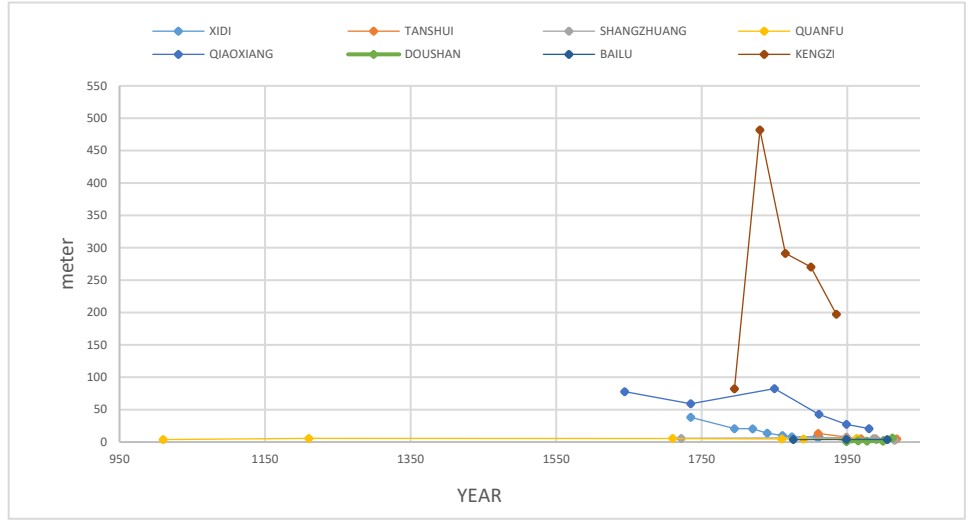

**Figure 12.** Variation diagram of ENN_SD.

**Table 4.** Test of the significance of correlation between ENN_MN and ENN_SD.

| | Shangzhuang | Doushan | Xidi | Bailu | Tanshui | Quanfu | Qiaoxiang | Kengzi |
|---|---|---|---|---|---|---|---|---|
| **Significance level-P (Valid: P < 0.05)** | 0.002 | 0.442 | 0.000 | 0.195 | 0.157 | 0.013 | 0.067 | 0.423 |
| **ENN_SD/ENN_MN** | 1.18 | – | 1.18 | – | – | 1.83 | – | – |

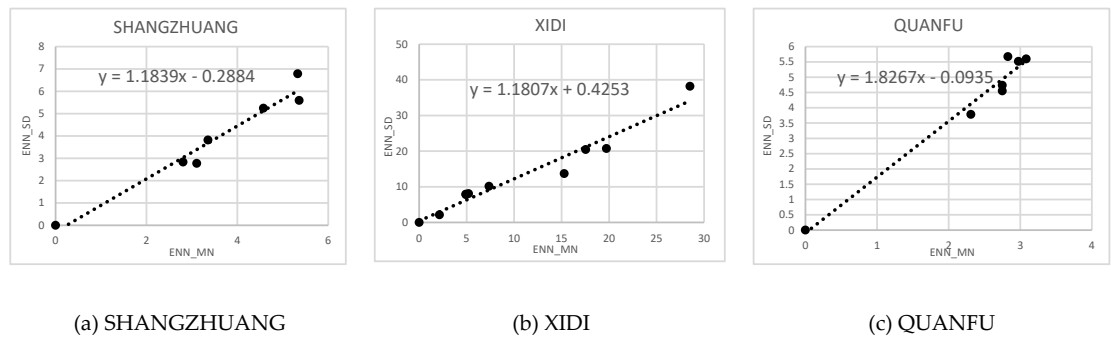

(a) SHANGZHUANG        (b) XIDI        (c) QUANFU

**Figure 13.** Linear regression for ENN_MN and ENN_SD.

(5) Patch density (PD).

This metric reflects the number of building patches distributed within a specific area (Figure 14).

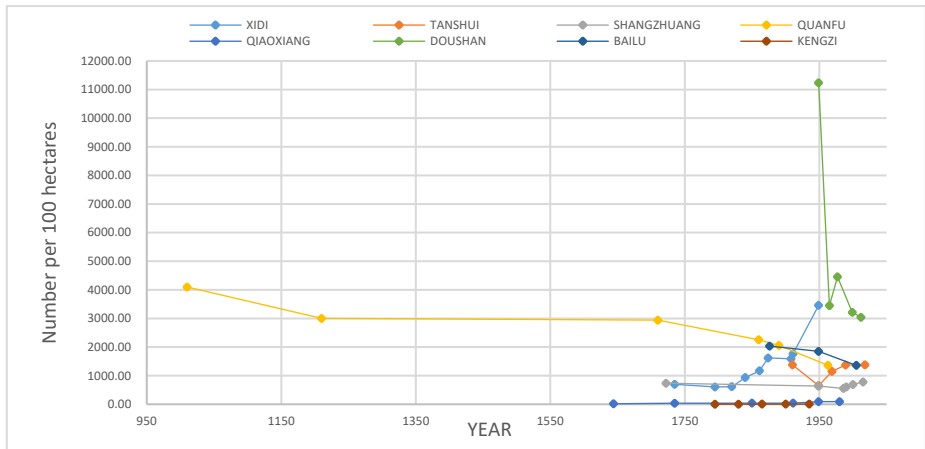

**Figure 14.** Variation diagram of patch density (PD).

There are three major kinds of cases:

For the first case, the growth value remained almost stable, fluctuating only slightly around a specific value. This situation occurred in Qiaoxiang, Kengzi, Shang Zhuang, and Tanshui, where spatial development was rarely interfered with by urban expansion and construction.

For the second case, there was an evident drop in decreased value. This situation occurred in Quanfu, Doushan, and Bailu. Of these, the patch density in Quanfu village remained relatively stable before the 18th century, and declined slowly after that. In Doushan and Bailu, the value began to decline in the late 19th century and the early 20th century.

Only Xidi village experienced the rare trend of a rapid increase in the PD value. That means an increasing number of people are located on a limited amount of land. This result indicated the restriction in the population flow in the valley region of Anhui province where Xidi was located.

3.2.2. The Data Changes of the Metrics at the Macro-Morphology Level

(1) Area within the boundary of village layout (AREA).

Figure 15 shows the changes in the data for all eight villages.

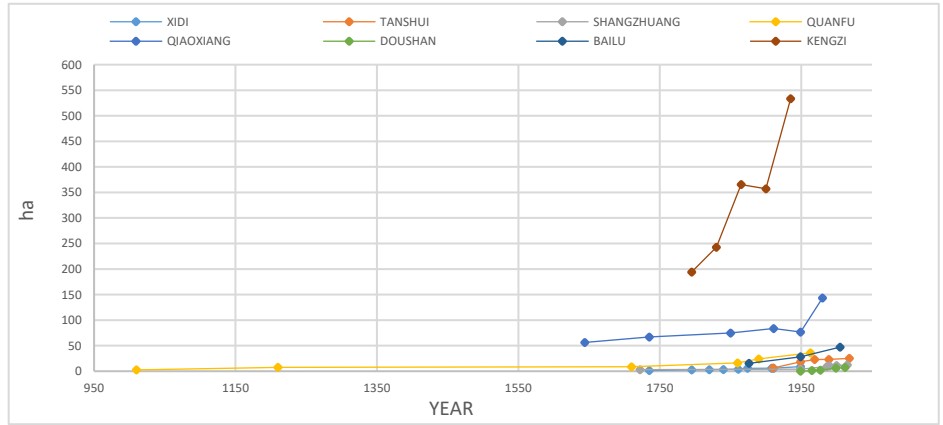

**Figure 15.** Variation diagram of AREA.

There were three main conditions associated with the development trends:

The first condition was a rapid increase in a straight line, represented by the data for Kengzi, Bailu, and Doushan (Figure 16a). Of these, Bailu and Doushan mainly experienced an edge-expansion mode, with a slower growth rate than Kengzi, which grew in the outlying mode.

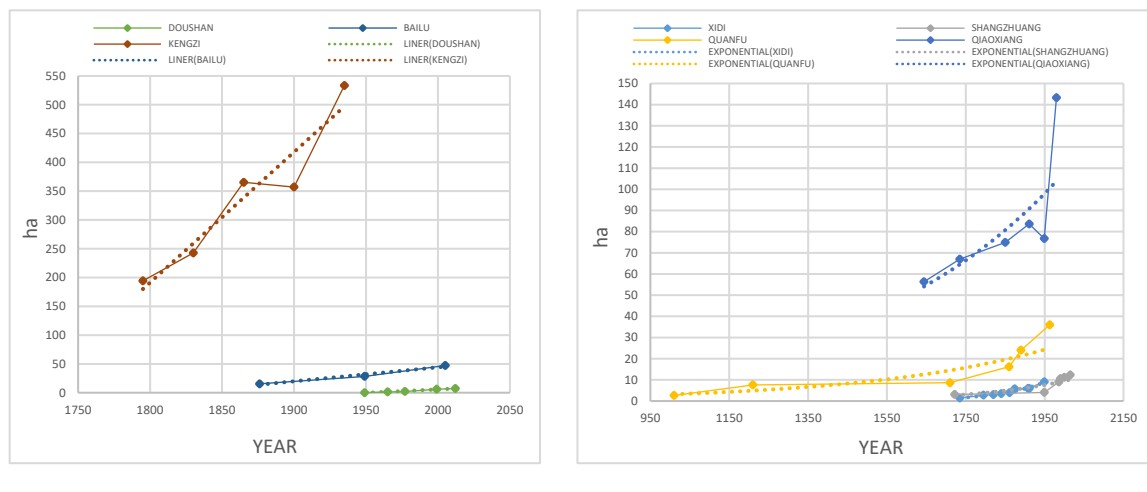

(**a**) Condition 1 - linear growth　　　　　　　　(**b**) Condition 2 - exponential growth

**Figure 16.** Trend analysis of AREA.

The second condition was an exponential increase at a slow speed, represented by the data for Quanfu, Xidi, Shangzhuang, and Qiaoxiang (Figure 16b). These villages did not have a large scale, and their spatial expansion was relatively comprehensive, either in a combined outlying and edge-expansion mode, or in an edge-expansion and infilling mode. In addition, those villages shared a common feature: they grew slowly in the early stage and rather rapidly in the latter. The turning point occurred around the time of the opium war and the early period after the founding of the PRC.

The third condition was when growth occurred in the shape of an S-curve. An example was seen in the data from Tanshui. This village expanded using both the infilling and outlying expansion modes. There may have been a threshold for spatial expansion established at the early stage of rural development (Figure 17).

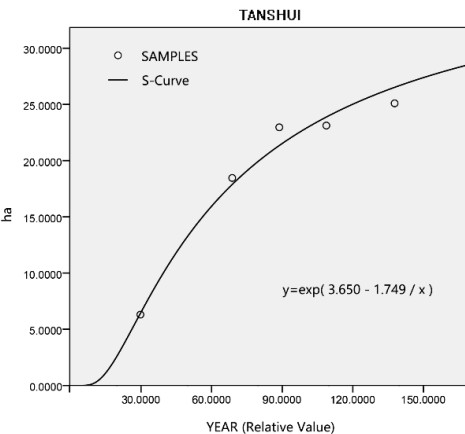

**Figure 17.** Condition 3: growth following an S-shaped curve.

(2) Fractal Dimension Index (FRAC)

The metrics established in this study, based on fractal theory, may reflect the complexity in village shapes when generated by self-organization [40].

Figure 18 shows that the value of each village experienced a complex trend of oscillation, with the amplitude of most changes in values attenuating with time. Regardless of the initial datum, the subsequent values consistently oscillated and approached the value of 1.12. This value of 1.12 is the average (calculated to two decimal places) of the fractal dimension index of the sample villages. Some samples were excluded in the calculation for the following reasons. First, the initial stage of the settlements was relatively unstable; therefore, the samples in this stage should be excluded. After that, for a certain village, if the retained samples in other stage were less than 5, the data were not suitable for the mean estimation; therefore, villages with a total sample size of less than 6 were not considered.

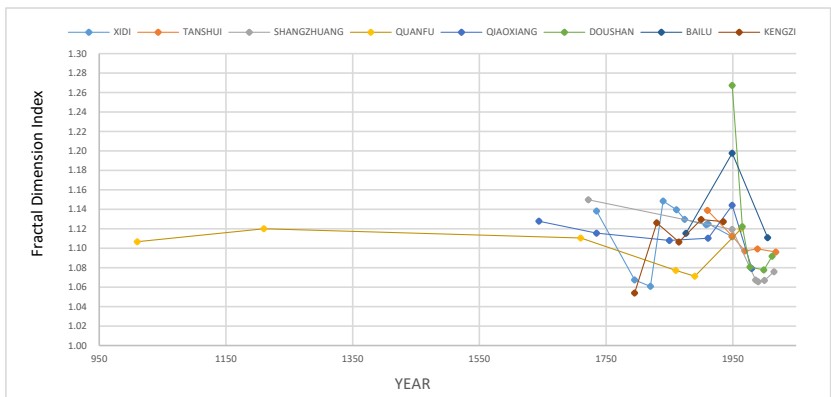

**Figure 18.** Variation diagram of FRAC.

The sample villages experienced some differences in the developing trend of metrics at the patch texture level. However, they demonstrated a common character with respect to the fractal feature of the village boundary. Villages may develop towards an approximately regular form, with a certain degree of complexity. The dimension of 1.12 may characterize a state of equilibrium, ultimately approached by many villages during their development.

## 4. Discussions and Conclusions

This study explored the spatial evolution characteristics of residential land in traditional villages in China, using multi-temporal morphological metrics covering the aspects of patch texture and macro-morphology. There was significant variability in the spatial patterns of different villages due to the unique natural and cultural environments in China [39]. The old rural social structure in China has

gradually disintegrated in the past 70 years; however, traditional settlements still somewhat reflect the spatial mapping of the historical social organization. All the study villages had been in place since the era of the peasant economy. Villages in different regions were under different governance modes.

In traditional China, the agricultural mode gradually transitioned from dry farming to irrigated agriculture from the north to the south; the importance of water conservancy cooperation and public defense also grew. Therefore, the basic cooperative unit of agriculture gradually changed from the family to the small kindred, and then to the large clan. Correspondingly, the population composition of a single village gradually changed from small families with multiple surnames to several family groups with several surnames, and then to a large family group with a single surname. When the large family group reached a certain size, it dispersed into several branches that became economically independent from each other. In other words, from Northern China to Southern China, villages gradually transitioned from mononuclear geographical settlements to polynuclear consanguineous settlements. This led to an increasing scale of the building patch area, the patch spacing, and the overall scale of a village.

Despite these differences, some common rules and trends were identified using the multi-temporal data of sample villages.

First, in respect to the continuity of spatial logic, spatial evolution characteristics have turned into a new direction since 1949, sometimes significantly fluctuating when faced with abnormal situations. Under the new social situation, the rural spatial development has occurred as a result of trial and error, without fully achieving a new balance.

Second, in respect to the stability of spatial texture, based on the change in the data for AREA_MN and ENN_MN, most small and medium-sized villages (with an area less than 30 ha) have maintained a relatively stable spatial texture during their development. In contrast, the spatial texture of large villages has significantly changed in a monotonic development trend, characterized by a decrease in building scale and an increase in the spatial distribution density. The correlation between the mean and the variance of patch area (AREA) and Euclidean nearest-neighbor distance (ENN) has not always been clear. No more than 40% of the sample villages had a stable coefficient of standard deviation (degree of variation) with respect to the separation distance and the building patch area during their spatial processes. The common mean value of the projected area of rural building patches was estimated at 100 $m^2$, with a value-float range of approximately 50–150 $m^2$. This can serve as a reference to establish the patch growth threshold in spatial simulations.

Third, in respect of the boundary of spatial expansion, when linking the expansion mode and the area growth of sample villages, villages with a single expansion mode generally appear out of control with respect to boundary development. By contrast, the blurred developable boundary may singularize the spatial expansion mode. If the developable boundary is determined, a village may combine outlying and infilling expansion modes. In addition, the fractal dimension of the overall spatial form enclosed by the boundary fluctuated and approached 1.12, which indicates an equilibrium level with respect to morphological evolution.

This study led to the following conclusions, which can serve as indispensable reference when simulating rural spatial dynamics and plans at the village scale:

(1)　In modern China, especially since the founding of the PRC, there has been a significant break in the stability of rural spatial development.

(2)　In comparison with large villages, most small and medium-sized villages have maintained a relatively stable spatial texture during their development.

(3)　The mean and variance of the patch area, and the Euclidean nearest-neighbor distance, are correlated in some cases.

(4)　The mode of rural expansion may be relevant to limitations in the total area of growth.

(5)　The complexity of the village boundary fluctuated and approached the equilibrium level in the process of village development.

(6)　The growth threshold of a single building patch may be about 100 $m^2$ and not more than 150 $m^2$ in general.

This study described some dynamic characteristics of the essential elements in settlement morphology in China. We believe these analysis and conclusions can inform future dynamic simulations and predictions for rural development.

**Author Contributions:** X.Y. conceived and designed the study. X.Y. and F.P. performed and verified the experiments. X.Y. and K.S. wrote, reviewed and edited the manuscript. All authors have read and agreed to the published version of the manuscript.

**Funding:** This research was funded by the National Natural Science Foundation of China, grant number 51908160, the Natural Science Foundation of Guangdong Province, China, grant number 2020A1515010681, the Natural Science Foundation of Shenzhen, China, grant number JCYJ20190806143403472, and the Fundamental Research Funds for the Central Universities, grant number 45001028.

**Conflicts of Interest:** The authors declare no conflict of interest.

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
