# Peer review of "Laws and Trends of the Evolution of Traditional Villages in Plane Pattern"

_sustainability, doi:10.3390/su12073005_

Round 1

Reviewer 1 Report

Dear Dr. Yang and Dr. Pu,

This is a robust and thoughtfully conceived study. It is well written and the methodology is well-elaborated and justified. You are to be congratulated on a thoughtful study.

In my view, this paper could be strengthened by addressing the perennial question of “so what”? The research shows different development patterns over time with variations between north-south. And yet, I am left wanting more of an explanation of why these distinct patterns have emerged, how they are linked to the distinct historical political economies and land use governance regimes and regulations. I would like to urge the authors to consider some of the contemporary relevance and policy implications of their work. I think this would expand the readership and offer compelling and relevant conclusions to this thoughtful study.

Author Response

Thank you for your comments concerning our manuscript entitled “Laws and Trends of the Evolution of Traditional Villages in Plane Pattern” (ID: sustainability-741240). Those comments were all valuable and very helpful for revising and improving our paper, as well as the significance of our findings. We have studied the comments carefully and have made corrections that we hope will be met with approval. Revised portions are in "Track Changes" mode in the manuscript. The main corrections in the paper and the responses to the reviewer’s comments are as follows:

Response to comment: Explaining why these distinct patterns have emerged, how they are linked to the distinct historical political economies and land use governance regimes and regulations, and considering some of the contemporary relevance and policy implications of their work.

Response: Thanks for the comments. The reviewer suggests that we provide the related social context that has shaped the distinct patterns, to provide readers with a better understanding of the history. In response, we have supplemented Section 4 by discussing the regionally differentiated social organizations that have supported different agricultural modes and public defense requirements (Lines 341~357 on Page 18). These have been the foundational factors influencing rural spatial characteristics. In addition we have deleted the seventh conclusion, given its similar meaning with part of the supplemental information.

Special thanks for your constructive comments.

Reviewer 2 Report

Introduction

Bibliographic research on landscape metrics should be updated.

Data and Methods

For each village the data and cartographic bases used and the period to which they refer must be clearly indicated.

Table 1 - Explain why scale of its smallest bounding rectangle is not the same for all villages

The graphic quality of figure 4 should be improved.

The text and Table 2 in sub-chapter 2.3 refer to the methodology and not to Data analysis. Title of this subchapter should be revised.

Figure 8 - standardize the decimal places of the figures (YY axis)

Results

The graphic quality of figure 5 should be improved.

Sub-chapter 3.2 refers to the methodology. Shouldn't be in the results chapter

Line 258 – 2.5 and 5.5, units are missing

Figure 17 – review: Negative area and negative years (??).What is the Case 3 that is mentioned in the legend of this figure?

Figure 18 - standardize the decimal places of the figures (YY axis)

YY axis - indicate legend

Line 320 - Explain why the dimension of 1.12 may characterize a state of equilibrium.

References

Review: authors must appear all - do not use et al. (Ex: Lines 407, 409, 423)

Author Response

Thank you for your comments concerning our manuscript entitled “Laws and Trends of the Evolution of Traditional Villages in Plane Pattern” (ID: sustainability-741240). Those comments were all valuable and very helpful for revising and improving our paper, as well as the significance of our findings. We have studied the comments carefully and have made corrections that we hope will be met with approval. Revised portions are in "Track Changes" mode in the manuscript. The main corrections in the paper and the responses to the reviewer’s comments are as follows:

  1. Response to comment: Bibliographic research on landscape metrics should be updated (in Section 1).

Response: In response to the reviewer’s suggestion, we have added references published in the last five years related to research on landscape metrics, particularly with respect to the application and improvement of landscape metrics in research on the dynamic spatial process (References NO. 13\19\20\22\26).

  1. Response to comment: For each village the data and cartographic bases used and the period to which they refer must be clearly indicated (in Section 2).

Response: We have improved this section in response to the reviewer’s suggestion. First, we added a brief description about data utilization and reconstruction based on relevant references (Line 139 ~ Line 143 on Page 5). Moreover, Figure 5 maps the processed spatial data, and the corresponding time nodes for each map have been added in this figure.

  1. Response to comment: Table 1 - Explain why scale of its smallest bounding rectangle is not the same for all villages (in Section 2).

Response: The smallest bounding rectangle represents the village scale. All the sample villages were in the same administrative level. The differences in scale for those villages reflect China's social reality. Explanations about this have been added with corresponding references (Line 145 ~ Line 149 on Page 5).

  1. Response to comment: The graphic quality of Figure 4 (in Section 2) and Figure 5 (in Section 3) should be improved.

Response: We have replaced the images with images with higher pixels, and adjusted the annotations in the images to make the meaning clearer (on Page 7 & 9).

  1. Response to comment: The text and Table 2 in sub-chapter 2.3 refer to the methodology and not to Data analysis. Title of this subchapter should be revised.

Response: We agree with this recommendation, and have modified the title as "Methods for spatial analysis" (Line 178 on Page 5).

  1. Response to comment: Figure 8 (in Section 2) and Figure 18 (in Section 3) - standardize the decimal places of the figures (YY axis),Figure 18 YY axis - indicate legend

Response: The decimal places have been standardized with two decimal places in response to the reviewer’s comments. A legend for the YY axis of Figure 18 has been added (on Page 11 & 17).

  1. Response to comment: Sub-chapter 3.2 refers to the methodology. Shouldn't be in the results chapter.

Response: Chapter 3.2 shows the results of spatial metric analysis corresponding to the analysis method presented in Chapter 2.3. Chapter 3.2.1 presents the calculation and regression analysis of the metric changes at the level of spatial texture. Chapter 3.2.2 is addresses the level of the integral shape. To clarify this intent, we renamed the title of Chapter 3 as "Results and analysis" (Line 197 on Page 9). The title of Chapter 3.2 was renamed as "metric data changes in the spatial sprawl"(Line 215 on Page 10).

  1. Response to comment: Line 258 – 2.5 and 5.5, units are missing.

Response: We have added the units for these data (Line 267 on Page 13).

  1. Response to comment: Figure 17 – review: Negative area and negative years (??). What is the Case 3 that is mentioned in the legend of this figure?

Response: We have excluded the domain and range that is without data displayed in the picture, and replaced the figure name with "Condition 3 – growth following an S-shaped curve". This figure refers to the 3rd Condition of rural area growth, discussed in lines 316~319. We also modified the name of Figure 16, which refers to the 1st and the 2nd Conditions, to connect it more clearly with the text (on Page 16).

  1. Response to comment: Line 320 - Explain why the dimension of 1.12 may characterize a state of equilibrium.

Response: Thanks for the comments. We have explained how the data were estimated and our analysis with respect to removing interfering data during the estimation (Line 325 ~ Line 331 on Page 17).

  1. Response to comment: Review: authors must appear all - do not use et al. (Ex: Lines 407, 409, 423)

Response: We have updated the references based on the reviewer’s comments (References NO. 15\16\18\20\21\23\26\30\32\37\39).

Special thanks for your helpful comments.